# Targeting Methionine Addiction of Osteosarcoma with Methionine Restriction to Overcome Drug Resistance: A New Paradigm for a Recalcitrant Disease

**DOI:** 10.3390/cancers17030506

**Published:** 2025-02-03

**Authors:** Yusuke Aoki, Yutaro Kubota, Noriyuki Masaki, Yasunori Tome, Michael Bouvet, Kotaro Nishida, Robert M. Hoffman

**Affiliations:** 1Department of Orthopedic Surgery, Graduate School of Medicine, University of the Ryukyus, 207 Uehara, Nishihara, Okinawa 903-0125, Japan; 2AntiCancer Inc., 7917 Ostrow St., Suite B, San Diego, CA 92111, USA; 3Department of Surgery, University of California, San Diego, 9300 Campus Point Drive #7220, La Jolla, San Diego, CA 92037, USA

**Keywords:** methionine addiction, Hoffman effect, methionine restriction, combination chemotherapy, drug resistance, osteosarcoma, recombinant methioninase, rMETase

## Abstract

Resistance to chemotherapy in osteosarcoma leads to poor prognosis, with a 5-year survival rate of approximately 20%; therefore, the development of novel therapies is required. Methionine addiction is a fundamental and general hallmark of cancer. Cancer cells need more exogenous methionine compared to normal cells due to their increased transmethylation. Methionine restriction targets methionine addiction and arrests cancer cells in the late-S/G2 phase. First-line chemotherapy for osteosarcoma, including methotrexate, doxorubicin, and cisplatinum, also targets cells in the S/G2 phase, resulting in synergy with methionine restriction to overcome drug resistance. The present review shows the synergistic efficacy of conventional chemotherapy and methionine restriction, and the potential of this new paradigm to overcome drug resistance of osteosarcoma is discussed.

## 1. Introduction

Osteosarcoma is the most common malignant bone tumor, often affecting children and young adults, although it is a rare disease. Multi-agent neoadjuvant and/or adjuvant chemotherapy, including methotrexate (MTX), doxorubicin (DOX), and cisplatinum (CDDP), has been administrated for osteosarcoma since the 1970s and improved the 5-year survival rate from less than 20% to around 60% [1,2,3,4,5,6,7]. However, if the osteosarcoma patient has a poor response to chemotherapy, the 5-year survival rate is approximately 20% [7,8,9,10], without improvement for over three decades.

Drug resistance in osteosarcoma is due to multiple mechanisms, which can be targeted to overcome the drug resistance. The overexpression of P-glycoprotein reduces intracellular drug accumulation in osteosarcoma cells, resulting in reduced chemotherapy efficacy. Nanoparticle drug delivery systems loaded with conventional chemotherapeutics improve intracellular drug accumulation and reduce systemic toxicity [11]. Wnt/β-catenin signaling promotes stem-like properties in osteosarcoma cells, leading to drug resistance. Combination therapies of conventional chemotherapies and Wnt/β-catenin inhibitors block stem-like properties, which cause drug resistance [12]. MicroRNAs regulate apoptosis, DNA repair, autophagy induction, and the promotion of cancer stem-cell properties in osteosarcoma cells, thus modulating drug resistance [13]. However, the development of novel, more effective therapeutic strategies for drug-resistant osteosarcoma is necessary to improve treatment outcomes.

Cancer cells can synthesize large amounts of methionine from homocysteine but are dependent on exogenous methionine [14,15,16]. We showed that cancer cells endogenously make normal or higher amounts of methionine from homocysteine, but the cancer cells still need large amounts of exogenous methionine in order to grow, due to increased transmethylation reactions [17,18,19,20,21,22]. This phenomenon is termed methionine addiction or the Hoffman effect [23,24], parallel to the glucose addiction of cancer, termed the Warburg effect [25]. Cancer cells need larger amounts of exogenous methionine than normal cells in order to grow, due to increased transmethylation reactions in cancer cells [17,19,22,26,27,28]. Wang et al. showed that tumor-initiating cells are highly methionine-addicted [29].

Osteosarcoma cells are inhibited by methionine restriction using recombinant methioninase (rMETase) [30,31,32,33]. We previously selected methionine-independent revertant 143B osteosarcoma cells from parental methionine-addicted 143B osteosarcoma cells by long-term culture in low-methionine media. The resulting methionine-independent revertant osteosarcoma cells displayed reduced cell migration, invasion, and proliferation capacity in vitro and loss of metastatic potential and tumor growth in vivo compared to their methionine-addicted parental cells [32]. These results indicate that osteosarcoma cells are addicted to methionine and that methionine addiction is linked to the malignancy of osteosarcoma cells. These results suggest that methionine addiction can be an effective therapeutic target for osteosarcoma cells.

We have also shown that methionine restriction selectively arrests cancer cells in the late-S/G_2_ phase of the cell cycle [34,35], which conventional cytotoxic chemotherapies target.

In the present review, we discuss the development of a therapeutic strategy for osteosarcoma, combining conventional chemotherapy and methionine restriction, including recombinant methioninase (rMETase), to overcome chemotherapy resistance in osteosarcoma.

## 2. Methionine Restriction Therapy for Osteosarcoma

Osteosarcoma cell lines, including 143B, MNNG-HOS, U-2OS, and Saos-2, are methionine-addicted and sensitive to rMETase [31]. rMETase significantly inhibited osteosarcoma cell growth in a dose-dependent manner in vitro and tumor volume in orthotopic xenograft nude mouse models compared to untreated controls [36]. The orthotopic xenograft nude mouse models were established by the following procedure: the mice were anesthetized via the subcutaneous injection of a cocktail [ketamine (20 mg/kg) (#11695-0702-1, Henry Schein, Inc., Melville, NY, USA), xylazine (15.2 mg/kg) (#59399-111-50, Akorn Operating Company LLC, Lake Forest, IL, USA), acepromazine maleate (0.48 mg/kg) (#0010-3827-01, Boehringer Ingelheim GmbH, Ingelheim, Germany)]. A skin incision was made on the left proximal tibia. A 1 mm diameter hole was made in the proximal part of the left tibia using a 5 mm blade (Medipoint Inc., Mineola, NY, USA). 143B osteosarcoma cells [2.0 × 10^5^ cells/5 mL phosphate-buffered saline and 5 mL Matrigel Matrix (#354234, Corning Inc., NY, USA)] were then injected through the hole using a 23-gauge needle. Treatment was initiated one week after cell injection and rMETase was injected intraperitoneally at 100 units/mouse daily for 21 days.

Genetically-engineered Salmonella, SGN1, which is capable of overexpressing L-methioninase and hydrolyzing methionine and thereby decreasing endogenous methionine and S-adenosyl-methionine (SAM), reduced the tumor growth and metastatic capacity and increased the survival of osteosarcoma subcutaneous-tumor and orthotopic metastatic mouse models, as well as patient-derived organoid and xenograft models, via a reduction in C1orf112 expression and mitochondrial functions [37].

## 3. Synergistic Efficacy of Conventional Chemotherapy and Methionine Restriction for Osteosarcoma

Methotrexate (MTX) inhibits dihydrofolate reductase (DHFR) [38], subsequently depleting 10-formyl-tetrahydrofolate and 5,10-methylente-tetrahydrofolate, which are essential for de novo purine synthesis and pyrimidine synthesis, respectively, and eventually reduces DNA synthesis. MTX also inhibits methionine S-adenosyl transferase (MAT), which is required for transmethylation reactions [39]. The combination of MTX and rMETase showed synergistic efficacy in an MTX-resistant osteosarcoma patient-derived orthotopic xenograft (PDOX) mouse model [40] (Table 1). The PDOX nude mouse models were established according to the following procedure: after induction of anesthesia as described above, a 1 mm diameter hole was made in the proximal part of the left tibia using a 5 mm blade (Medipoint Inc., Mineola, NY, USA). A 1 mm^3^ tumor fragment of osteosarcoma obtained from a subcutaneous tumor in a PDX mouse model was inserted into the hole.

Cisplatinum (CDDP) binds to DNA, subsequently interfering with DNA replication [41]. CDDP combined with rMETase showed significant efficacy in osteosarcoma PDOX mouse models [42,43,44] (Table 1).

Docetaxel (DOC) affects the M-phase of the cell cycle by interfering with microtubules [45]. Cancer cells that escape the blocking of the S/G2-phase of the cell cycle induced by methionine restriction are then killed by DOC [46]. DOC showed synergistic efficacy with AG-270, an inhibitor of methionine adenosyltransferase 2α (MAT2A), which is involved in transmethylation reactions and methionine addiction in cancer cells [47]. The combination therapy of DOC and rMETase showed synergistic efficacy in osteosarcoma PDOX mouse models [48] (Table 1). rMETase also reversed acquired DOC resistance in osteosarcoma cells, which was established by culturing the parental 143B cells in increasing DOC concentrations (0.14–24 nM) over 5 months [49] (Table 1).

Azacytidine (AZA) decreases cytosine methylation in DNA by inhibiting DNA methyltransferase [50]. Methionine restriction decreases S-adenosylmethionine (SAM) [19], which is the only methyl-donor for DNA, RNA, and histone transmethylation reactions. In osteosarcoma PDOX mouse models, the combination of AZA and rMETase showed synergistic efficacy [51] (Table 1).

Rapamycin inhibits cell growth by interfering with mammalian target of rapamycin (mTOR) signaling, which is regulated by the PI3K/AKT pathway [52,53,54,55]. Rapamycin has shown efficacy in osteosarcoma cell lines (5). The synergistic efficacy of rapamycin plus oral rMETase (o-rMETase) was reported in mammary gland osteosarcoma PDOX mouse models [56] (Table 1).

Recently, we showed the synergistic efficacy of ethionine and rMETase in an osteosarcoma cell line and not normal cells [57] (Table 1). Ethionine is a structural analog of methionine, with a methyl group replaced by an ethyl group, and is an anti-metabolite of methionine. Cells synthesize S-adenosylethionine (SAE) from ethionine, instead of S-adenosylmethionine (SAM), which is an essential methyl-donor for DNA, RNA, and histone, as stated above [58]. The synergistic efficacy of the combination of ethionine and rMETase on osteosarcoma cell viability may be due to the mimetic effect of ethionine on methionine and the depletion of exogenous methionine by rMETase.

Clinical administration of oral rMETase (o-rMETase) to human patients, as an oral dietary supplement, has been used alone and in combination with conventional chemotherapy in various types of cancer. o-rMETase has shown promising efficacy in patients with invasive lobular breast cancer, prostate cancer, ovarian cancer, pancreatic cancer, brain cancer, and rectal cancer [59,60,61,62,63,64,65,66,67,68]. Blood levels of PSA, CA19–9, and CEA measured in prostate, pancreatic, breast, and rectal cancers decreased continuously with the administration of o-rMETase.

Although o-rMETase has not been administrated to osteosarcoma patients yet, potential efficacy can be expected, as in other types of cancers.

**Table 1 cancers-17-00506-t001:** Synergistic efficacy of methionine restriction and chemotherapy in osteosarcoma.

Chemotherapy	Methionine Restriction	Study Model	Result	Reference
Methotrexate	o-rMETase	Osteosarcoma of pelvis, PDOX mouse model	Significant efficacy of the combination of o-rMETase and MTX; no significant efficacy of MTX alone	Aoki Y et al. [40]
Cisplatinum	ip-rMETase	Recurrent CDDP-resistant metastatic osteosarcoma of femur, PDOX mouse model	Significant efficacy of the combination of o-rMETase and CDDP; no significant efficacy of CDDP alone	Igarashi K et al. [42]
Cisplatinum	o-rMETase	Osteosarcoma of pelvis, PDOX mouse model	Significant efficacy of the combination of o-rMETase and CDDP; no significant efficacy of CDDP alone	Higuchi T et al. [43]
Cisplatinum	o-rMETase	Osteosarcoma of mammary gland, PDOX mouse model	Significant efficacy of the combination of o-rMETase and CDDP at 3.0 mg/kg; comparable to CDDP alone at 6.0 mg/kg	Masaki N et al. [44]
Docetaxel	o-rMETase	Osteosarcoma, PDOX mouse model	Significant efficacy of the combination of o-rMETase and DOC; no significant efficacy of DOC alone	Aoki Y et al. [48]
Docetaxel	rMETase	DOX-resistant 143B osteosarcoma cells in vitro	Synergestic efficacy of the combination of o-rMETase and DOX; no significant efficacy of DOX alone	Morinaga S et al. [49]
Azacytidine	o-rMETase	Osteosarcoma of pelvis, PDOX mouse model	Significant efficacy of the combination of o-rMETase and AZA; no significant efficacy of AZA or DOX alone	Higuchi T et al. [51]
Rapamycin	o-rMETase	Osteosarcoma of mammary gland, PDOX mouse model	Significant efficacy of the combination of o-rMETase and rapamycin;	Masaki N et al. [56]
Ethionine	rMETase	143B osteosarcoma cells and Hs27 fibroblast cells in vitro	Synergestic efficacy of the combination of o-rMETase and ethionine and down regulation of c-MYC in osteosarcoma cells; no significant efficacy in fibroblast cells	Aoki Y et al. [57]

rMETase: recombinant methioninase; o-rMETase: oral rMETase; PDOX: patient-derived orthotopic xenograft; CDDP: cisplatinum; DOC: docetaxel; DOX: doxorubicin; AZA: azacytidine.

## 4. Discussion

The methionine dependence of cancer was originally shown by Sugimura et al. in 1959, studying tumors in rats. Deleting methionine from the diet arrested tumor growth more than depleting other amino acids [14]. Hoffman and Erbe showed that cancers are methionine-addicted in 1976 [17]. Wang et al. also showed that tumor-initiating cells are highly methionine-addicted [29]. All cancers tested are addicted to methionine [69]. The present review reports the single and synergetic efficacy of methionine restriction, including rMETase, in combination with conventional anticancer therapies in osteosarcoma cells in vitro and in mouse models.

MTX, DOX, and CDDP, which are first-line chemotherapies for osteosarcoma targeting cells in the S/G2-phase, where cancer cells are arrested by methionine restriction, including rMETase [34,35]. MTX also targets folate metabolism by inhibiting DHFR, which is necessary for endogenous methionine synthesis metabolism, and therefore decreases endogenous methionine synthesis. Methionine-addicted cancer cells are also addicted to folate [18], possibly contributing to the synergistic efficacy of rMETase and MTX [40].

Small Phase I and II clinical trials have been carried out to determine the toxicity of a methionine-restricted diet in cancer patients without critical adverse events [70,71,72]. However, a methionine-restricted diet is unacceptable to some patients. We previously conducted a pilot Phase I clinical trial of methionine restriction by means of intravenous rMETase infusion, in which no clinical toxicity was observed in any patient [73]. As mentioned above, oral rMETase (o-rMETase) has shown clinical promise alone and in combination with conventional chemotherapies for breast cancer [64,68], prostate cancer [59,60,61], ovarian cancer [59], pancreatic cancer [63], brain cancer [67], and rectal cancer [62,66], with no adverse events reported.

We have shown that methionine restriction, including rMETase, can reverse drug resistance in osteosarcoma [40,42,43,44,48,49,51,56,57], suggesting a future new paradigm to overcome drug resistance in osteosarcoma, which has been a recalcitrant problem for over three decades.

## 5. Conclusions

Methionine restriction, including rMETase, is synergistic in combination with conventional chemotherapy, which is a future new paradigm to overcome drug resistance in osteosarcoma. Methionine restriction therapy is promising for drug-resistant osteosarcoma due to methionine addiction. Further clinical studies are necessary.

## Data Availability

The data presented in this study are available in PubMed at https://pubmed.ncbi.nlm.nih.gov.

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
