# Peer review of "Targeting Methionine Addiction of Osteosarcoma with Methionine Restriction to Overcome Drug Resistance: A New Paradigm for a Recalcitrant Disease"

_cancers, 2025, doi:10.3390/cancers17030506_

Round 1
Reviewer 1 Report
Comments and Suggestions for Authors
Dear Editor-in-Chief
Thank you for inviting me to review the manuscript titled “Targeting methionine addiction of osteosarcoma with methionine restriction to overcome drug resistance: A new paradigm for a recalcitrant disease”. In this manuscript, the authors demonstrated methionine addiction, the Hoffman effect, and the potential of methionine restriction therapy in overcoming chemotherapy resistance in osteosarcoma. However, further clarity is required.
1. In the Introduction, the relationship between methionine addiction and osteosarcoma progression is presented abruptly. I suggest bridging this gap with some examples or whatever.
2. There are overlapping contents in some parts of the manuscript. For example, sections "Synergistic Efficacy" and section "Potential Clinical Strategy" overlap in content.
3. Major parts of the article are separated by numerous paragraphs, which could distract the readers. To increase the integrity of the article, I suggest merging some sections together.
4. Section "4. Potential clinical strategy of oral-rMETase" lacks enough in-depth data. I recommend merging it with the previous section.
5. I recommend discussing the potential challenges such as toxicity in the Discussion.
Comments on the Quality of English LanguageSome minor grammatical and language mistakes are present.
Author Response
1. In the Introduction, the relationship between methionine addiction and osteosarcoma progression is presented abruptly. I suggest bridging this gap with some examples or whatever.
Authors response: Thank you for your suggestion. The relationship between methionine addiction and osteosarcoma progression is presented with some results from previous reports in the revised version.
2. There are overlapping contents in some parts of the manuscript. For example, sections "Synergistic Efficacy" and section "Potential Clinical Strategy" overlap in content.
Authors response: Thank you for pointing this out. Section "4. Potential clinical strategy of oral-rMETase" is merged with the previous section, “Synergistic efficacy of conventional chemotherapy and methionine restriction for osteosarcoma” to reduce overlapping contents in the revised version.
3. Major parts of the article are separated by numerous paragraphs, which could distract the readers. To increase the integrity of the article, I suggest merging some sections together.
Authors response: Thank you for your suggestion. Section "4. Potential clinical strategy of oral-rMETase" is merged with the previous section, “Synergistic efficacy of conventional chemotherapy and methionine restriction for osteosarcoma” in the revised version.
4. Section "4. Potential clinical strategy of oral-rMETase" lacks enough in-depth data. I recommend merging it with the previous section.
Authors response: Thank you for your recommendation. Section "4. Potential clinical strategy of oral-rMETase" is merged with the previous section, “Synergistic efficacy of conventional chemotherapy and methionine restriction for osteosarcoma” in the revised version.
5. I recommend discussing the potential challenges such as toxicity in the Discussion.
Authors response: Thank you for your recommendation. The toxicity of methionine restriction therapy is discussed in the Discussion section in the revised version.
Reviewer 2 Report
Comments and Suggestions for Authors
In this manuscript, Aoki et al. highlighted the synergistic potential of methionine restriction, particularly using recombinant methioninase, with conventional chemotherapy in overcoming drug resistance in osteosarcoma. Their findings underscore a promising therapeutic paradigm to enhance treatment efficacy in this recalcitrant disease. I have few queries before its consideration for publication:
1) I feel that the content of this review article appears too limited to be considered publishable. I recommend including general information about osteosarcoma, current treatment modalities, treatment outcomes, and the challenges associated with drug resistance. This would help make the article more comprehensive and appealing to a wider readership.
2) Please draw a good graphical abstract.
3) The abstract contains repetitive details about methionine addiction. Streamline the content to avoid redundancy and improve readability.
4) The introduction lacks a clear explanation of the Hoffman Effect. Adding a concise description would enhance comprehension for a broader audience.
5) While discussing rMETase efficacy, the methodology section does not clearly describe experimental setups for the xenograft models. Provide more detail on procedures to ensure reproducibility.
6) Table 1 lack sufficient detail. Improvise it.
7) The conclusion is overly brief and does not address potential limitations or future research directions. Expand this section to provide a balanced summary.
8) Terms such as "methionine addiction" and "methionine restriction" are used inconsistently. Ensure consistent terminology throughout the manuscript.
9) Improve the English language.
Comments on the Quality of English LanguageMinor improvements required.
Author Response
1) I feel that the content of this review article appears too limited to be considered publishable. I recommend including general information about osteosarcoma, current treatment modalities, treatment outcomes, and the challenges associated with drug resistance. This would help make the article more comprehensive and appealing to a wider readership.
Authors response: Thank you for pointing this out. Although general information, current treatment modalities, and treatment outcomes are already explained in the Introduction, the challenges associated with drug resistance are added in the Introduction section in the revised version.
2) Please draw a good graphical abstract.
Authors response: Thank you for your suggestion. We drew a graphical abstract.
3) The abstract contains repetitive details about methionine addiction. Streamline the content to avoid redundancy and improve readability.
Authors response: Thank you for pointing this out. The section of “Methionine addiction and restriction therapy for osteosarcoma was changed to “Methionine restriction therapy for osteosarcoma” and repetitive details about methionine addiction was eliminated from the section, which is addressed in the revised version.
4) The introduction lacks a clear explanation of the Hoffman Effect. Adding a concise description would enhance comprehension for a broader audience.
Authors response: Thank you for pointing this out. Explanation of the Hoffman effect is added to the Introduction section in the revised version.
5) While discussing rMETase efficacy, the methodology section does not clearly describe experimental setups for the xenograft models. Provide more detail on procedures to ensure reproducibility.
Authors response: Thank you for pointing this out. The experimental setups for the xenograft nude mouse models is clearly described in the revised version.
6) Table 1 lack sufficient detail. Improvise it.
Authors response: Thank you for pointing this out. Table 1 was improved in the revised version.
7) The conclusion is overly brief and does not address potential limitations or future research directions. Expand this section to provide a balanced summary.
Authors response: Thank you for pointing this out. Potential limitations, such as toxicity and acceptance of methionine restriction therapy is added in the Discussion and Conclusions section in the revised version.
8) Terms such as "methionine addiction" and "methionine restriction" are used inconsistently. Ensure consistent terminology throughout the manuscript.
Authors response: Thank you for pointing this out. Methionine addiction is hallmark of cancer cells, including osteosarcoma, and can be targeted as a therapeutic strategy by methionine restriction. The section of “Methionine addiction and restriction therapy for osteosarcoma was changed to “Methionine restriction therapy for osteosarcoma” to avoid confusion.
9) Improve the English language.
Authors response: Thank you for pointing this out. English is improved in the revised version.
Reviewer 3 Report
Comments and Suggestions for Authors
It is a quite short albeit comprehensive review on the role of methionine depletion in osteosarcomia. Paper is very interesting and well written and may be published as it is.
There are some small editorial errors, which could be corrected upon proof-reading. The are as follows:
1./ page 2, line 59: should be effect not Effect;
2./ all the names od organisms shouldbe in italics;
3./ page 3, line 115: should be donor not doner.
Author Response
1./ page 2, line 59: should be effect not Effect;
Authors response: Thank you for pointing this out. Effect has been corrected to effect in the revised version.
2./ all the names od organisms should be in italics;
Authors response: Thank you for pointing this out. All the names od organism, including Salmonella and SGN1, are in italics in the revised version.
3./ page 3, line 115: should be donor not doner.
Authors response: Thank you for pointing this out. Donor has been corrected to doner in the revised version.
Round 2
Reviewer 1 Report
Comments and Suggestions for Authors
Dear Editor-in-Chief,
The Authors addressed my issues satisfactorily; therefore, I recommend the publication of the manuscript in its present form.
Reviewer 2 Report
Comments and Suggestions for Authors
The authors have addressed all my queries, however, the similarity index seems to be bit high (54%). Kindly reduce it.
Comments on the Quality of English LanguageMinor improvements required.